# Novel Antiviral Molecules against Ebola Virus Infection

**DOI:** 10.3390/ijms241914791

**Published:** 2023-09-30

**Authors:** Mila Collados Rodríguez, Patrick Maillard, Alexandra Journeaux, Anastassia V. Komarova, Valérie Najburg, Raul-Yusef Sanchez David, Olivier Helynck, Mingzhe Guo, Jin Zhong, Sylvain Baize, Frédéric Tangy, Yves Jacob, Hélène Munier-Lehmann, Eliane F. Meurs

**Affiliations:** 1School of Infection & Immunity (SII), College of Medical, Veterinary and Life Sciences (MVLS), Sir Michael Stoker Building, MRC-University of Glasgow Centre for Virus Research (CVR), Glasgow G61 1QH, UK; 2Unité Hépacivirus et Immunité Innée, CNRS, UMR 3569, Département de Virologie, Institut Pasteur, 75015 Paris, France; maillardpatrick55@gmail.com (P.M.); eliane.meurs@pasteur.fr (E.F.M.); 3Unit of Biology of Emerging Viral Infections, Institut Pasteur, 69007 Lyon, France; alexandra.journeaux@pasteur.fr (A.J.); sylvain.baize@pasteur.fr (S.B.); 4Interactomics, RNA and Immunity Laboratory, Institut Pasteur, 75015 Paris, France; anastasia.komarova@pasteur.fr; 5Unité de Génomique Virale et Vaccination, Institut Pasteur, 75015 Paris, France; valerie.najburg@pasteur.fr (V.N.); r.y.sanchezdavid@qmul.ac.uk (R.-Y.S.D.); frederic.tangy@pasteur.fr (F.T.); 6Université Paris Cité, 75013 Paris, France; yves.jacob@ti-ar-Jacob.com; 7Blizard Institute—Faculty of Medicine and Dentistry, Queen Mary University of London, London E1 2AT, UK; 8Unité de Chimie et Biocatalyse, CNRS, UMR 3523, Institut Pasteur, Université de Paris, 75015 Paris, France; olivier.helynck@pasteur.fr (O.H.); helene.munier-lehmann@pasteur.fr (H.M.-L.); 9CAS Key Laboratory of Molecular Virology and Immunology, Unit of Viral Hepatitis, Shanghai Institute of Immunity and Infection, Center for Biosafety Mega-Science, Chinese Academy of Sciences, Shanghai 200023, China; finn.sloan@gmail.com (M.G.); jzhong@ips.ac.cn (J.Z.); 10Unité Génétique Moléculaire des Virus à ARN, CNRS, UMR 3569, Département de Virologie, Institut Pasteur, 75015 Paris, France

**Keywords:** Ebola virus, measles virus, VP35, PKR, PACT, RIG-I, drug screen

## Abstract

Infection with Ebola virus (EBOV) is responsible for hemorrhagic fever in humans with a high mortality rate. Combined efforts of prevention and therapeutic intervention are required to tackle highly variable RNA viruses, whose infections often lead to outbreaks. Here, we have screened the 2P2I_3D_ chemical library using a nanoluciferase-based protein complementation assay (NPCA) and isolated two compounds that disrupt the interaction of the EBOV protein fragment VP35IID with the N-terminus of the dsRNA-binding proteins PKR and PACT, involved in IFN response and/or intrinsic immunity, respectively. The two compounds inhibited EBOV infection in cell culture as well as infection by measles virus (MV) independently of IFN induction. Consequently, we propose that the compounds are antiviral by restoring intrinsic immunity driven by PACT. Given that PACT is highly conserved across mammals, our data support further testing of the compounds in other species, as well as against other negative-sense RNA viruses.

## 1. Introduction

Ebola virus (EBOV) remains a public health concern, since periodic outbreaks occur in Africa and several cases have spread into other continents (see [1] for a recent review). In humans, EBOV causes a cytokine storm leading to hemorrhagic fever, which is responsible for its high lethality rate [2]. Current FDA-approved treatments are monoclonal antibodies only evaluated for efficacy against the Zaire EBOV strain (https://www.cdc.gov/vhf/ebola/treatment/index.html, accessed on 21 September 2023). Therefore, complementary treatments with broad action are required for this rapidly evolving RNA virus [3].

EBOV is an enveloped non-segmented negative-sense single-stranded RNA ((-)ssRNA) filovirus that contains seven genes arranged from 3′ to 5′ encoding the nucleoprotein (NP), viral proteins VP35, VP40, VP24, VP30, glycoprotein (GP), and the RNA-dependent RNA polymerase [4]. The structure and crosstalk between these proteins have been uncovered through numerous studies aiming at identifying druggable targets [5,6,7,8,9,10]. One of the EBOV proteins in the spotlight is VP35, since it is a cofactor of the viral RNA-polymerase complex required for viral replication [5,7,10]. The VP35 structure comprises an N-terminus domain (NTD, aa 180) followed by an oligomerization domain (OD, aa 80–145) and a C-terminus domain (CTD, aa 215–340). VP35 CTD has been also termed an interferon (IFN) inhibitory domain (IID) because of its implication in counteracting multiple antiviral responses; RNA-based viruses such as EBOV tend to generate secondary structures consisting of double-stranded RNA (dsRNA) which are immunogenic, that is, can be detected by a variety of dsRNA intracellular sensors to trigger an immune response to clear the invading RNA [11]. However, VP35IID binding to viral dsRNA masks its recognition by host dsRNA-sensors to avoid their driven IFN-α/β production [12,13]. More in detail, VP35IID contains a key hydrophobic residue (F239) whose sidechain contacts dsRNA blunt ends, as well as a central basic patch (CBP) where amino acid (aa) residues R312 and R322 provide a charged surface for dsRNA binding, while R305, K309 and K319 increase dsRNA interaction strength [14]. Both main VP35 functions (polymerase cofactor and dsRNA binding) have been targeted in silico [15,16,17,18]. 

In parallel, VP35IID interacts directly with dsRNA-dependent protein kinase (PKR) through paired combinations of CBP residues R305/K309/R312, preventing its activation to bypass its role in translation arrest, subsequently allowing viral translation to occur [19,20,21]. Moreover, VP35IID binding to a PKR activator (PACT) [22] prevents its interaction with and activation of retinoic acid-inducible gene I (RIG-I), further impairing IFN production [12,23], as well as the PACT-mediated activation of RNAi [22,24]. The PKR and PACT complex interplay with other RIG-I-like Receptors (RLRs), as MDA5 and LGP2 are less characterized [25], especially in the context of EBOV infection.

Since VP35IID interaction with PKR and PACT impairs their antiviral functions, we hypothesized that releasing PKR and PACT from VP35IID sequestration may restore their antiviral potential, facilitating viral clearance. Accordingly, our goal in this study was to screen for compounds disrupting such interactions. To do so, we set up an in cellula nanoluciferase protein complementation assay (NPCA, [26]) to screen a pre-filtered library of chemical compounds [27,28]. The efficacy of the identified compounds as antiviral agents against EBOV and measles virus was addressed, as was their effect on the intrinsic and innate immune response.

## 2. Results

### 2.1. Generation and Assessment of a VP35IID/NtPKR and VP35IID/PACT NPCA

In order to identify inhibitors of the interaction between VP35IID and PKR or PACT, we established an in cellula nanoluciferase protein complementation assay (NPCA). For this, we constructed plasmids expressing VP35IID, the N-terminus of PKR (NtPKR; aa 1-265) as well as a full-length PACT (Figure 1a). In addition, to enhance the specificity of this assay, we generated plasmids individually expressing each of their double-stranded RNA-binding domains (DRBDs) abbreviated as K1 and K2 for PKR, and A1 and A2 for PACT. In the case of PACT, we also generated constructs with A1 and A2 together (NtPACT), and A2 with M3 (A2M3), where M3 represents the CTD of PACT (Figure 1a). Each construct was generated in all possible combinations, namely, in fusion with each luciferase moiety inserted into either the N- or C-terminus (Nt or Ct) of each protein. For each assay, reconstituted luciferase activity resulting from the protein interactions was measured 24 h after co-transfection in HEK293T cells. NtPKR/PACT homo- or heterodimerization served as positive controls, given their known ability to interact through their DRBDs [29]. Since the strongest luciferase signal was obtained when the luciferase moieties 1 and 2 were fused to the N-terminus of NtPKR or PACT (N1) and VP35IID (N2), respectively (Appendix A), we used these construct combinations subsequently.

We observed that the luciferase signal was higher when VP35IID interacted with NtPKR than with PACT (Figure 1b), presumably because individual DRBDs from PKR (K1 and K2) interact qualitatively more strongly with VP35IID than the ones from PACT (A1 and A2) (Figure 1b). Therefore, the NtPACT and A2M3 constructs of PACT were generated to determine whether the M3 region was somehow masking PACT DRBDs, which would account for the lower luciferase signal in the VP35IID/PACT pair than the VP35IID/NtPKR one. However, the luciferase signal resulting from the VP35IID/NtPACT or VP35IID/A2M3 interaction was weaker than that obtained when VP35IID interacts with full-length PACT (Figure 1c). Thus, full-length PACT and the two DRBDs of PKR (NtPKR) were required for the strongest interaction with VP35IID. 

To further assess the specificity of the VP35IID interactions with NtPKR or PACT by NPCA, as well as to ensure that this system could display a detectible drop in luciferase signal when protein–protein interactions are compromised, we generated a panel of VP35IID constructs with relevant aa substitutions to alanine (Figure 1d). The interaction of VP35IID with NtPKR was significantly reduced with the K309A/R312A (*p*-value of 0.0433) and qualitatively decreased in R305A/R312A VP35IID, confirming the role of these residues in the PKR/VP35 interaction [20]. In contrast, only a qualitative decrease could be detected for the interaction with PACT. These results confirm the reported importance of the VP35IID central basic patch in the interactions with PKR and PACT, their interaction strength, and give an indication of the luc signal decrease to expect from inhibitory compounds.

In conclusion, since the strongest luciferase signals were obtained when VP35IID was co-expressed with NtPKR or full-length PACT all tagged in their N-terminus, we proceeded to use this NPCA as a platform towards screening compounds which would abrogate these interactions.

### 2.2. Identification of Compounds Targeting the Interaction of VP35IID with PACT or NtPKR

To set up the screening platform abrogating protein interactions between VP35IID and NtPKR or PACT, HEK293T cells were transfected with the plasmid N2-VP35IID and either the plasmids N1-NtPKR or N1-PACT. After 24 h, the cells were trypsinated and distributed into 96-well-plates containing the 2P2I_3D_ library of 1664 compounds, each at a final concentration of 20 μM [27], as described in Materials and Methods. Luciferase activity measured after another 24 h yielded 287 hits for VP35IID/NtPKR and 262 hits for VP35IID/PACT that could inhibit the luciferase activity by 60–90%. False positives due to direct interference with the luciferase enzymatic activity and/or to cytotoxicity were then discarded through two additional screenings, as well as non-specific hits known to be frequently selected whatever the screening procedure. From these three rounds of screening and analysis (Figure 2a), 44 compounds confirmed their ability to: (i) inhibit luciferase activity to less than 1%, (ii) reduce cell viability to less than 1% and (iii) decrease the VP35IID/PACT and/or VP35IID/NtPKR split luc signal in a 60–90% range. Out of these 44 compounds, 41 were available from the suppliers and were further tested at four different concentrations (1, 5, 10 and 20 μM) on HEK293T cells 24 h post-transfection with each pair of plasmids (VP35IID/NtPKR or VP35IID/PACT) (Appendix A). Compounds **13** and **36** (reference K221-3357/MolPort-007-903-447 (ChemDiv) and STK283971/MolPort-002-995-533 (Vitas-M Laboratory), respectively, (Appendix A) were the most efficient in inhibiting the interaction of VP35IID with either NtPKR or PACT (Appendix A). Therefore, they were selected for further study.

To ensure that the compounds were not cytotoxic, an additional MTT assay was performed in another cell line (Huh7.25/CD81) upon 24 h and 72 h treatment. After 24 h, cell viability was preserved for both compounds, while at 72 h only a dose-response toxicity was observed for compound **36** at concentrations around 7 µM; compound **13** rather appeared to be cytostatic, since cell viability did not decrease from 30% even above 10 µM (Appendix A).

### 2.3. Compounds **13** and **36** Decrease EBOV RNA and Infectious Virus

Next, we tested the effect of either compound at 0, 1, 5 and 10 µM on two cell types (HEK293T and Huh7), uninfected or infected with EBOV (MOI 0.1) for 24, 48 and 72 h. Both cell lines were assessed for viability through the experiment through a different assay (CellTiter-Glo) (Appendix A); again, cell viability was generally not affected by addition of the compounds, except for compound **13** at 10 µM at 3 dpi in both cell types, independently of EBOV infection. 

The effect of the compounds on EBOV was determined by quantifying the amount of intracellular (internal) and extracellular (external) viral RNA at 24 (Day 1) and 72 (Day 3) hours post infection (hpi), respectively, as well as production of infectious virus released in the supernatant at these time points (EBOV titer) (Figure 3a). We evidenced that the presence of both intracellular and extracellular EBOV RNA was reduced by the compounds in a dose-dependent manner already at 24 hpi (Figure 3b), and the production of the infectious virus (titer) qualitatively decreased between 0.5–1.5 logs (Figure 3c).

Given the reported ability of VP35 to coat and mask viral dsRNA from dsRNA sensors to prevent IFN induction [14], we wanted to assess whether impairing VP25IID from interacting with two dsRNA sensors (PKR and PACT) would have an impact on IFN induction. Of note, we believe that most of the abrogation of IFN induction by VP35 is precisely through its dsRNA-binding capability, which is not directly targeted with this NPCA; we designed the NPCA screen to abrogate the interaction between VP35IID with NtPKR or PACT to unlock their antiviral effect against EBOV evidenced in the previous section. Therefore, as expected, upon assessing the RNA expression of the two early (IFNβ, IFNα1) and one late (IFNα2) IFN genes 24 hpi, IFNs were not induced by EBOV infection (Appendix A), nor altered by the compounds. 

### 2.4. Compounds **13** and **36** also Impair MV Independently of IFN

Given that dsRNA sensors are at the frontline defense against RNA virus infections, these proteins are common targets of RNA viruses (reviewed in [11]). Therefore, we next investigated whether the above anti-EBOV compounds could be antiviral against another (-)ssRNA virus. For this, we used measles virus (MV, family *Paramyxoviridae*), harboring or not (MVΔV) the non-structural virulence V protein [30]. In agreement with the above observations, the two compounds reduced the intracellular presence of the two MV variants in a dose-dependent manner, as shown by RT-qPCR and immunoblot analysis through decreased expression of the nucleoprotein (N) (Figure 4a) independently of IFNβ transcription (Figure 4b). Moreover, PKR and PACT total protein levels remained constant throughout the experiments, suggesting that they could be available to perform other functions. Altogether, these data show that compounds **13** and **36** are antiviral agents, at least against negative RNA viruses such as EBOV and MV, and that their effect is independent of the IFN induction pathway.

### 2.5. Mechanism of Action of Compounds ***13*** and ***36***

We then examined the effect of compounds **13** and **36** on combinatorial associations of the RIG-I/PACT/PKR proteins using NPCA. We corroborated the ability of RIG-I to interact with PACT and NtPKR [31,32], and we showed that VP35IID can also interact with RIG-I. Moreover, all interactions were dose-response-inhibited by compound **13** and with less efficacy by compound **36** (Figure 5a). Thus, in addition to interfering with the interaction of VP35IID with PACT or NtPKR, the compounds may interfere with interactions of PACT or NtPKR with other RLRs as RIG-I.

We have previously reported that the association of PKR and PACT is involved in the induction of pro-inflammatory cytokines in response to stress, and this association could be inhibited at the level of their DRBDs by luteolin, a natural compound member of the flavonoid family [33], reviewed in [34]. We therefore compared the action of the compounds **13** and **36** to that of luteolin and observed that they could also inhibit the NtPKR/PACT interaction, as well as their interaction with VP35IID (Figure 5b). Therefore, it is tempting to speculate that by interfering with protein interactions involved in EBOV or MV infections, as well as in IFN and pro-inflammatory cytokine induction, compounds **13** and **36** behave as antiviral agents while controlling the inflammatory response often associated with these viral infections.

## 3. Discussion

In this study, we have screened a chemical library of 1664 compounds and identified two compounds disrupting the interaction of the EBOV protein fragment VP35IID with the cellular proteins PACT and NtPKR. K221-3357 and STK283971 (referred here as compounds **13** and **36**, respectively) have the ability to inhibit negative-sense RNA viruses from different families (EBOV from *Filoviridae* and MV from *Paramyxoviridae*); therefore, these compounds may have broad spectrum antiviral activity. Determining their antiviral extent individually or in combination against other negative-sense RNA viruses such as Crimean–Congo hemorrhagic fever, influenza or rabies viruses, may offer alternative treatments in case of emergency. Both compounds appeared to be nevertheless specific against negative-sense RNA viruses, as we examined their effect on a positive-sense RNA virus as HCV, where no antiviral effect was found (not shown). 

Since negative-sense RNA viruses have to be transcribed to a positive-sense RNA to translate viral proteins, this process generates dsRNA intermediates unique to negative-sense RNA viruses that may be detected by dsRNA sensors and/or by dsRNA RNAses as Dicer [35]; VP35 is known to interfere with this process [24,36]. Interestingly, the interaction of VP35IID with either NtPKR or PACT was two orders of magnitude stronger than with itself (Figure 1b), suggesting that PKR and PACT may interfere with VP35 di-/tri-/tetramerization required to act as polymerase cofactor [5] and/or efficiently coat dsRNA [14]. However, the interaction strengths may be different with full-length proteins. Of note, we have presented evidence that VP35IID interactions with NtPKR or PACT can take place in the absence of viral dsRNA, suggesting for the first time that their functionality may be compromised irrespective of dsRNA binding. This is of importance because other screens [15] have focused on abrogating the specific interaction between VP35 and dsRNA, which would restore IFN induction, presumably enhancing a systemic cytokine storm. Given that the compounds were selected through a screen platform aiming to disrupt protein–protein interactions, instead of abrogating the dsRNA binding by VP35IID as in [15,37,38], the compounds may not interfere with the dsRNA-binding capabilities of VP35IID, NtPKR or PACT. Thus, from our studies, we do not know unequivocally whether the compounds also abrogated dsRNA binding to these proteins; therefore, further investigation is warranted. Detailed high-resolution microscopy studies are also required to determine whether the compounds interfere with VP35 viral polymerase cofactor. 

Additionally, we observed a weak inhibitory effect of the compounds on the interaction between NtPKR or PACT with RIG-I (Figure 5a). Of note, this is the first report showing a direct interaction between PKR and RIG-I, although they have been previously reported to be in close vicinity in virus-induced stress granules or in complexes [31,32,39]. Although the significance of this interaction and of its inhibition by the compounds remains to be further explored, inhibiting the interaction between PACT and RIG-I suggests that IFN synthesis driven by RIG-I would not occur in presence of the compounds. In fact, we evidenced the antiviral effect of the compounds to occur independently of IFN levels during EBOV and MV infections. Therefore, we classify compounds **13** and **36** as boosters of intrinsic immunity. In our proposed model (Figure 6), the two compounds inhibit protein interactions, which may make dsRNA more accessible to host RNAses. Complementarily, since PACT can use any of its dsRNA domains to interact with Dicer [35], its known participation on RNA interference (RNAi) may also decrease viral RNA levels. Further studies are required to determine whether key interactions of PACT with RNAi pathway components, such as TRBP or DICER1, are not disrupted by the compounds, and that RNAi pathway functionality is indeed restored by the compounds during infection. Additionally, the central role of PACT in the mechanism of action of the compounds could be further confirmed by testing the compounds in PACT-deficient backgrounds. In line with this, we would like to emphasize that PACT is a highly conserved protein across several mammalian species relevant to EBOV infection in comparison to PKR (Appendix A). Also, in contrast to PKR and RLRs, PACT and other RNAi components are not ISGs in a range of vertebrate species (Appendix A), which suggests a translation of our results without species barriers. In view of the current evidence, the functional significance of PACT interactions with IFN-inducible RLR will have to be reassessed. Thereby, we endorse these compounds for animal testing, especially because their lack of effect on IFN induction may not aggravate a typical EBOV systemic storm, avoiding further morbidity and mortality in humans. Studies assessing a synergistic effect when the compounds are administered in combination with current monoclonal treatments are also desirable. In addition, we have evidenced luteolin to target combinations of VP35IID, NtPKR or PACT heterodimers, offering further mechanistic explanations to its previously reported antiviral effect [40,41,42,43,44,45].

Our screen also identified compounds increasing the interaction of VP35IID with both NtPKR and PACT (#2, MolPort-005-948-404, NAT17-347111 (AnalytiCon Discovery, GmbH); #9, MolPort-001-572-901, 7,961,700 (ChemBridge Corporation); #16, MolPort-010-967-340, K783-5489 (ChemBridge Corporation); #23, MolPort-002-527-771, STOCK1N-55055 (InterBioScreen Ltd., New York, NY, USA); #29, MolPort-002-611-996, STL343551 (Vitas-M Laboratory, Ltd., Champaign, IL, USA); #35, MolPort-019-950-511, STOCK1N-77683 (InterBioScreen), Appendix A); therefore, it is tempting to speculate that these compounds may fill the dsRNA pocket between VP35IID and NtPKR or PACT in our NPCA, strengthening their interaction when viral dsRNA is absent. Nevertheless, these compounds were not investigated in the context of an infection because our objective was to find and test compounds inhibiting these proteins’ interactions to impair viral infection. 

According to the viability tests we performed in different cell lines, ≤10 µM treatment for 24 h seems to give an antiviral effect without compromising cell viability, while lower concentrations (e.g., ≤5 µM) should be used within 72 h. These orders of magnitude are comparable to drugs already used to inhibit EBOV [46]. Although our viability tests may offer a starting point to choose a concentration range for experimentation, we want to remark that the compounds were pulsed only once and then kept through the assay, without pulsing each day. Therefore, the viability should be re-assessed when translating these results into animal tests, since the compounds’ half-life in plasma and cytotoxicity in multicellular organisms remains to be determined.

Overall, we identified two compounds and validated their antiviral action against two negative-sense RNA viruses (EBOV and MV) in two cell types also tested for cell viability. Finally, we propose a model of action of the compounds through PACT-driven RNAi and highlight further areas of research.

## 4. Materials and Methods

### 4.1. Cell Culture

HEK293T, Huh7 and Huh7.25/CD81 cells were cultured in Dulbecco’s modified Eagle’s Medium (DMEM + GlutaMAX; Gibco laboratories; Grand Island, NY, USA) supplemented with 10% heat-inactivated fetal bovine serum (FBS) (Hyclone; GE Healthcare Life Sciences, Cranbury, NJ, USA), 1% nonessential amino acids (Gibco), 1000 U/mL penicillin and 0.1 mg/mL streptomycin (Invitrogen/ThermoFisher, Carlsbad, CA, USA, #15140122) at 37 °C with 5% CO_2_. Vero (CRL1586) cells were cultured as above but supplemented with 5% heat-inactivated fetal bovine serum (FBS) instead.

### 4.2. Antibodies

Anti-PACT polyclonal antibodies were purchased from Santa Cruz Biotechnology (#sc-377103). Mouse monoclonal antibodies anti-Myc and rabbit polyclonal anti-FLAG antibodies were from Santa Cruz. Mouse monoclonal anti-β-actin antibody was from Sigma (Irvine, UK; #A1978). Goat anti-rabbit IgG (H+L) Secondary Antibody DyLight 800 4X PEG (#SA5-35571) and Goat anti-mouse IgG (H+L) secondary Antibody DyLight 680 (#35518) were from Invitrogen. The anti-measles nucleoprotein mouse monoclonal antibody (#ab9397) was from Abcam (Cambridge, UK).

### 4.3. Expression Vectors

The Open Reading Frame (ORF) sequences corresponding to N-terminus PKR 1-265 [47] and full-length PACT [29] were cloned by in vitro recombination into pDONR207 (Gateway^®^ BP Clonase™ II Enzyme MIX, Invitrogen™ by Life Technologies™) and transferred by LR recombination into vectors expressing in-frame complementary fragments of nanoluciferase, either at their N- (luciferase N1 or N2) or C- (luciferase C1 or C2) terminus [26,48]. A similar procedure was used to generate the different fragments of PKR (K1, K2) and PACT (A1, A2, A2-M3) in frame with the luciferase moieties (Appendix A). Because of French regulatory constraints on highly pathogenic micro-organisms to which EBOV belongs, only manipulation of constructs of ≤500 nucleotides (nt) are accepted in laboratories without special authorization. Therefore, the codon-optimized version of EBOV VP35 (GenBank: AF086833.2, Ebola Virus-Mayinga, Zaire, 1976) generated in [49] was used to clone the EBOV VP35IID fragment of 378 nt (215–340 aa): aaagcccgacattagtgctaaggacctgcgcaacatcatgtacgatcacctgccaggctttggcaccgcctttcaccagctggtgcaggtcatctgcaagctgggcaaagactccaattctctggacatcatccacgccgagttccaggcttccctggccgagggcgattcaccccagtgcgctctgatccagatcaccaagagggtgcccattttccaggatgcagctccccctgtgatccacattcgctccaggggcgacatccccagggcttgccagaagtccctgcgaccagtccctccctccccaaagatcgacaggggctgggtctgcgtgtttcagctgcaggacggcaagaccctgggtctgaagatttga) using the VP35F/R primer pair into Zero Blunt^®^ TOPO^®^ (ThermoFisher, USA, #K283020); this sequence served as a PCR template for generating all VP35IID-related nanoluciferase moieties constructs using primers without and with a stop codon consisting of the primer VP35-B1 paired with either VP35-B2N or VP35-B2S, respectively (Appendix A). The PCR products were purified and cloned into pDONR207 by BP reaction to obtain the pDONR207+VP35IID+NoStop and pDONR207+VP35IID+stop constructs, respectively. The VP35IID amino acid changes K319A/R322A (M1), R312 (M2), F239 (M3), R305A/K309A (M4), K309A/R312A (M5) and R305A/R312A (M6) were generated by site-directed mutagenesis (QuikChange II Site-directed mutagenesis kit, Agilent) with the primer pairs M1F/M1R, M2F/M2R, M3F/M3R, M4F/M4R, M5F/M5R and M6F/M6R, respectively, using WT VP35IID in pDONR207 as a template and transferring the resulting mutants into N2 destination vector by Gateway LR recombination. The pDONR207 vector containing the RIG-I sequence was prepared as described [48] and transferred to vectors expressing in frame the nanoluciferase moiety N2 at their N-t.

### 4.4. Nanoluc Protein Complementation Assay

The nanoluciferase protein complementation assay (NPCA) was performed as described [26,33]. Briefly, HEK293T cells (32,000 cells/well were seeded in 96-well opaque white plates. After 24 h, cells were transfected using PEI (linear polyethylenimine PEI “MAX”, Polysciences Inc., Warrington, Philadelphia, PA, USA, #23966) with 200 ng of the VP35IID, NtPKR or PACT constructs in fusion with the luciferase moieties. At 24 h post-transfection, the cells were incubated in the presence of 40 μL of Nano-Glo luciferase reagent (Promega, Southampton, UK; #N1120). Luciferase enzymatic activity was measured with a Centro XS LB960 luminometer (Berthold, Bad Wildbad, Germany) using 100 ms integration time unless otherwise indicated. Luciferase (Luc) signal given by the luminometer is plotted as absolute Luc signal.

### 4.5. Compound Library Screening with NPCA

The 2P2I_3D_ chemical library of 1664 compounds dedicated to orthosteric modulation of protein–protein interactions was acquired from Protisvalor (Marseille, France; purchased via MolPort, Latvia) and stored in 384-well plates. This library contains protein–protein interaction (PPI)-like compounds corresponding to medicinally important privileged structures identified as core structures in numerous therapeutics, after filtering of 8.3 million compounds representing the main chemical providers commercially available [28,50,51]. The screening was performed on a Freedom EVO^®^ platform (Tecan, Paris, France). One µL of each compound (20 mM in DMSO) from the 2P2I_3D_ chemical library was spiked into white, flat-bottom, bar-coded tissue culture 96-well plates (Greiner, Courtaboeuf, France). For each plate, the first four wells of column 1 and the four last wells of column 12 contained only 1 µL of DMSO to act as positive controls of fluorescence signal (cells co-transfected either with the VP35IID/NtPKR pair or the VP35IID/PACT pair). The remaining wells of columns 1 and 12 also contained 1 µL of DMSO, but the added cells were co-transfected with vectors expressing both luciferase moieties (Luc1 or Luc2) to serve as negative controls. HEK293T cells were seeded in three 10 cm^2^ round plates and two 484 cm^2^ square plates (Corning) to have 8.2 × 10^6^ and 40 × 10^6^ cells/plate, respectively. Then, 24 h after seeding, cells of the square plates were co-transfected with a total amount of 125 µg of either N1-PACT or N1-NtPKR with N2-VP35IID while cells of two 10 cm^2^ plates were co-transfected with 25.6 µg of the vectors expressing only Luc1 and Luc2. On the following day, the cells were trypsinated, washed with DMEM and resuspended at 3.2 × 10^5^ cells/mL in DMEM with 10% FBS and without antibiotics to load 100 μL/well into the plates containing 1 μL of each compound at 2 mM, so the final concentration was 20 μM. After 24 h, 40 μL of Nano-Glo luciferase reagent (Promega #N1120) were added to measure the luciferase activity on a M1000 Pro (Tecan) using a 100 ms integration time. For each plate, positive and negative controls were used to calculate the Z’-factor [51] following the formula Z’ = 1 − 3 × (σ^+^ + σ^−^)/(μ^+^ − μ^−^), where σ^+^ and σ^−^ correspond to standard deviations for positive and negative controls, respectively, and μ^+^ and μ^−^ correspond to means of luminescence signal measured for positive and negative controls, respectively.

### 4.6. Cell Viability Assays

In parallel to the drug screens, a methyl-thiazol-tetrazolium (MTT) cell proliferation assay (Sigma, CGD-1) was performed following the manufacturer’s instructions. Briefly, a non-transfected preparation of HEK293T cells was distributed into a further set of 96-well opaque white plates containing the compounds from the 2P2I^3D^ chemical library. The first column only contained 1% DMSO (no compound) serving as positive controls, and the last column was devoid of cells, DMSO and compounds, to serve as a blank, to ensure that the substrate does not give any signal by itself. After 24 h incubation at 37 °C, the MTT solution was added, and the formation of crystals was monitored. Once they started to appear in the positive control, the MTT solvent was added, and the crystals dissolved by pipetting up and down with the Freedom EVO^®^ platform (Tecan) robot; the signal was immediately measured spectrophotometrically at 570 nm. 

Further MTT viability assays were conducted in Huh7.25-CD81. Then, 1 × 10^4^ cells seeded in 96-well culture plates were incubated for 24 h to reach 60–70% confluence. After incubation for 24 h to reach 60–70% confluence, the medium was replaced by 200 µL of medium containing the compounds 13 or 36 at 1, 5, 10, 20, 30 and 40 µM or DMSO at equivalent concentrations. At 24 h or 72 h post treatment, supernatants were replaced by 100 µL OPTIMEM containing 10% MTT and incubated at 37 °C during 4 h. Then, supernatants were replaced by 100 µL of lysis buffer/well and incubated for another 15 min under orbital agitation according to the manufacturer’s instructions (Sigma); plates were read at 570 nm. IC_50_ estimations were conducted on GraPhad Prism 2.0 upon fitting a non-linear curve. The percentage of viable cells was presented as a percentage of the untreated samples at each time point for each sample.

### 4.7. Whole Luciferase Enzymatic Assay

A further set of the 2P2I^3D^ chemical library was used to discard false positives resulting from inhibition of an in vitro transcribed whole luciferase enzymatic activity in 96-well opaque white plates. As above, the first column only contained the enzyme in 1% DMSO (no compound) serving as positive control, while the last column was devoid of enzyme to serve as negative control.

### 4.8. Real-Time Quantitative PCR and Analysis

For EBOV infection and IFN analysis, total cellular RNA was extracted on day one, according to the instructions of the RNeasy kit manufacturer (Qiagen, Hilden, Germany). Potential genomic DNA contamination was minimized by treatment with DNase I (RNase-free DNase set; Qiagen). Reverse transcription was used to generate cDNA from extracted RNA, SuperScript III reverse transcriptase, oligo (dT), a deoxynucleoside triphosphate (dNTP) mixture, RNaseOUT, dithiothreitol (DTT), and 5× RT buffer (all from Invitrogen). cDNA was then quantified, using GAPDH as a reference, by quantitative PCR (qPCR) on an LightCycler480 (Roche) using TaqMan Universal Master Mix and commercial TaqMan primers and probes for GAPDH (VIC probe). Other TaqMan assays were carried out with the primers and probes listed in Appendix A. Genomic DNA contamination was checked by testing for amplification in RNA samples without the reverse transcriptase step. The results were normalized to the amount of GAPDH cDNA and are plotted as RNA relative expression.

For MV RNA presence, total cellular RNA was extracted using TRI-Reagent (Sigma), according to the manufacturer’s instruction and reverse-transcribed using 1 µg of total RNA with either OligodT18 (ThermoFisher) or MV genome-specific primers ([30], Appendix A). Host cell mRNA transcription and MV RNA genome levels were quantified by a two-step quantitative real-time PCR (RT-qPCR) assay. RT-qPCR was performed using an AbiPrism 7900HT machine, with a FastStart Universal SYBR Green Master (Roche, Welwyn Garden City, UK). The results were normalized to the amount of GAPDH cDNA and are plotted as RNA relative expression.

### 4.9. Immunoblot

Cells were washed once with PBS and scraped into CHAPS buffer (50 mM Tris-HCl pH 7.5, 140 mM NaCl, 5 mM EDTA, 5% glycerol, 1% CHAPS) containing cOmplete™ EDTA-free protease inhibitor and PhosSTOP™ phosphatase inhibitor (Roche). Protein concentrations were determined according to Bradford using Protein Assay Dye Reagent Concentrate (Bio-Rad, Hercules, CA, USA, #5000006). Protein electrophoresis was performed in NuPAGE 4–12% Bis-TRIS gels (Invitrogen, Waltham, MA, USA). After protein blotting, nitrocellulose membranes (Fisher Bioblock Scientific, Illkirch, France) were blocked with 2.5% skimmed milk and probed with specific antibodies, as described in the figure legends. An Odyssey scanner was used for immunoblot image acquisition with the Odyssey 3.1 software (Li-Cor Biosciences, Lincoln, NE, USA).

### 4.10. Viruses

The Zaire strain of EBOV was obtained after two passages in Vero E6 cells using the plasma from a fatally infected patient during an outbreak in Gabon in 2001. The titer of the viral stock was 5 × 10^6^ focus-forming Units/mL with confirmed absence of mycoplasma. Infections in experiments were conducted at MOI of 0.1 and subsequent titer was determined by immunostaining of infectious foci on Vero cells with mouse polyclonal primary antibodies from ascites conjugated to anti-mouse alkaline phosphatase.

Recombinant MV vaccine Schwarz strain has been previously described [52], as well as MVΔV [30,48]. For experiments, HEK293T cells were seeded in 24-well plates at 2 × 10^5^ cells/well. After 24 h, cells were left uninfected or infected with MV or MVΔV at MOI of 0.3 for 60 min (adsorption).

### 4.11. Statistical Analysis

All experiments were independently repeated at least twice with triplicate samples, unless otherwise stated. Plotted results are shown as mean ± standard deviation (SD). Differences between more than two groups were assessed by the Kruskal–Wallis test, since sample distributions did not follow a Gaussian distribution. Statistical significance is indicated with asterisks, where one represents a *p*-value < 0.05, two a *p*-value < 0.01, three a *p*-value < 0.001, and four a *p*-value < 0.0001. Although all graphs were assessed for statistical significance, only significant comparisons are indicated, for clarity.

## Figures and Tables

**Figure 1 ijms-24-14791-f001:**
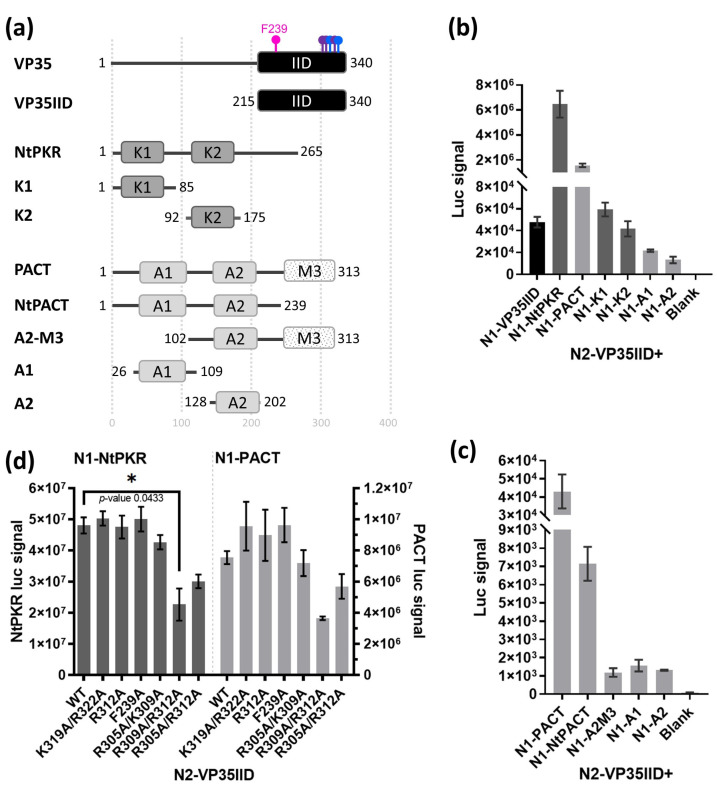
Interaction of VP35IID with NtPKR or PACT individual domains in cellula. (**a**) Schematic representation of Zaire EBOV VP35 and VP35IID constructs with approximate position of amino acid (aa) changes used in (**d**) indicated with pins, as well as the N-terminus (Nt) aa 1-265 fragment of PKR (NtPKR), full-length PACT and the derived constructs for either PKR or PACT, indicating the position of the two double-stranded RNA binding domains (DRBDs: K1 and K2 for PKR and A1 and A2 for PACT, respectively), and the M3 domain of PACT. (**b**) HEK293T cells (30,000 cells/well in 96-well plates) were co-transfected with 100 ng of the VP35IID construct with the luciferase moiety 2 at its N-terminus (N2-VP35IID) in the presence of each of the following constructs with the luciferase moiety 1 at their N-terminus: NtPKR, PACT and each of their individual DRBDs. (**c**) N2-VP35IID construct transfected in presence of full-length PACT or its specific domains depicted in (a) (NtPACT, A2M3, A1 or A2), each tagged with the luciferase moiety 1 at their N-terminus. (**d**) Cells were co-transfected with either N1-NtPKR or N1-PACT in the presence of each of the VP35IID, WT or constructs with the indicated aa substitutions. (**c**,**d**) experiments were performed as in (**b**); luciferase enzymatic activity was measured for 100 ms (**b**,**c**) or 5 s (**d**). Representative graphs are shown; only statistically significant differences are indicated. Blank shows very low levels of signal coming from the plastic plate without cells, media, DMSO or compounds to ensure the luciferase substrate does not give any signal by itself in comparison with the transfected samples. * *p* < 0.05.

**Figure 2 ijms-24-14791-f002:**
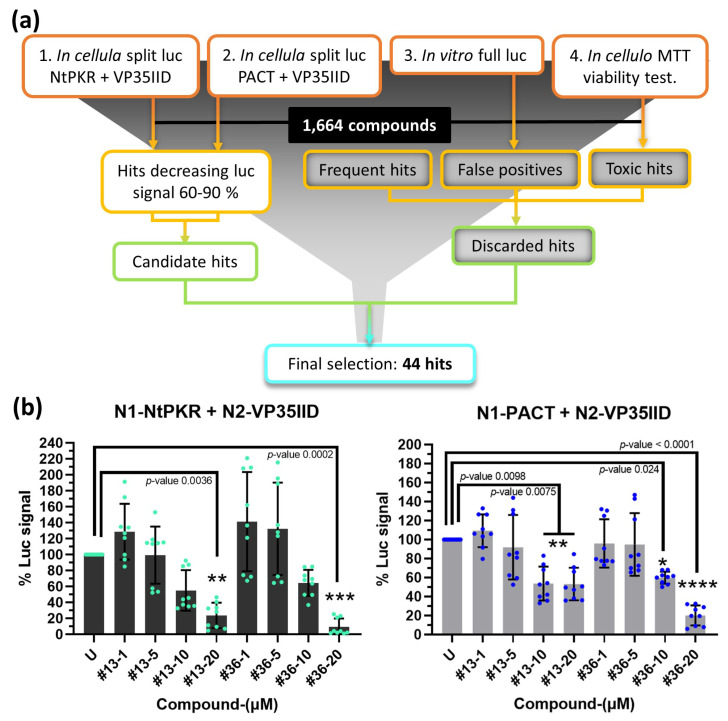
Selection and testing of compounds. (**a**) High-throughput screening flow-through diagram of a chemical library using NPCA. The 1664 compounds were applied to the split luc in cellula interactions of VP35IID/PACT (1) and VP35IID/NtPKR (2), full-length luciferase in vitro (full luc, 3) and to untransfected cells to perform an MTT viability test (4). Hits from (1) and (2) giving a 60–90% luc signal decrease comprised the first selection of hits from which false positives (3: hits also inhibiting full luc < 1%), toxic hits (4: hits reducing cell viability < 1%) and frequent hits were subtracted. The final selection resulted in 44 hits. (**b**) Disruption of the interaction NtPKR/VP35IID and PACT/VP35IID by compounds **13** and **36** extracted from Appendix A; compilation of three independent experiments with three experimental replicates (in cyan and blue dots, respectively) normalized to the untreated (U) and expressed in percentage to appreciate the effect of increasing concentrations of compounds **13** and **36** (1, 5, 10 and 20 µM) on the interaction NtPKR/VP35IID (left) or PACT/VP35IID (right). U, untreated with DMSO or compounds. Only statistically significant comparisons are indicated, for clarity. Both compounds significantly inhibit both protein interactions at 20 µM after 24 h (*p*-values 0.0036, 0.0002, 0.0098 and <0.0001, respectively), and they are also significantly effective at 10 µM against PACT/VP35IID interaction (*p*-values 0.0075 and 0.024, respectively). Statistical significance is indicated with asterisks, where * represents a *p*-value < 0.05, ** a *p*-value < 0.01, *** a *p*-value < 0.001, and **** a *p*-value < 0.0001.

**Figure 3 ijms-24-14791-f003:**
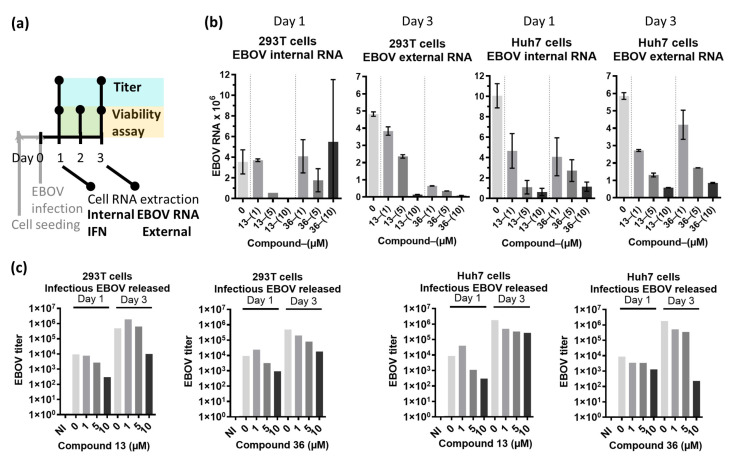
Effect of compounds **13** and **36** in EBOV infection. (**a**) Experimental procedure diagram. HEK293T or Huh7 cells were seeded in 12-well or 96-well plates at 4 × 10^6^ cells/plate for HEK293T cells or 3 × 10^6^ cells/plate for Huh7 cells. After 24 h, they were infected with EBOV for 1 h at 37 °C. After adsorption, the medium was replaced with culture medium containing compound **13** or **36** at a final concentrations of 1, 5 or 10 µM; an aliquot of medium was taken at this moment representing time 0 of infection (Day 0). Aliquots of the supernatants were taken at day 1, 2 and 3 for viability assay by CellTiter-Glo (Appendix A), and at day 1 and day 3 for titration of the virus released from the cells (infectious EBOV released). RNA was extracted after 24 h (Day 1) for measuring of endogenous EBOV RNA (EBOV RNA internal) and for expression of cytokines (Appendix A), and after 72 h (Day 3) for measuring of the EBOV RNA secreted by the cells (EBOV RNA external). (**b**) Internal and external EBOV RNA from infected 293T or Huh7 cells untreated or treated with a concentration range of compounds **13** or **36**; all scales are ×10^6^. Graphs display averages of 2 technical replicates. (**c**) EBOV titer estimated by immunostaining; NI, not infected sample.

**Figure 4 ijms-24-14791-f004:**
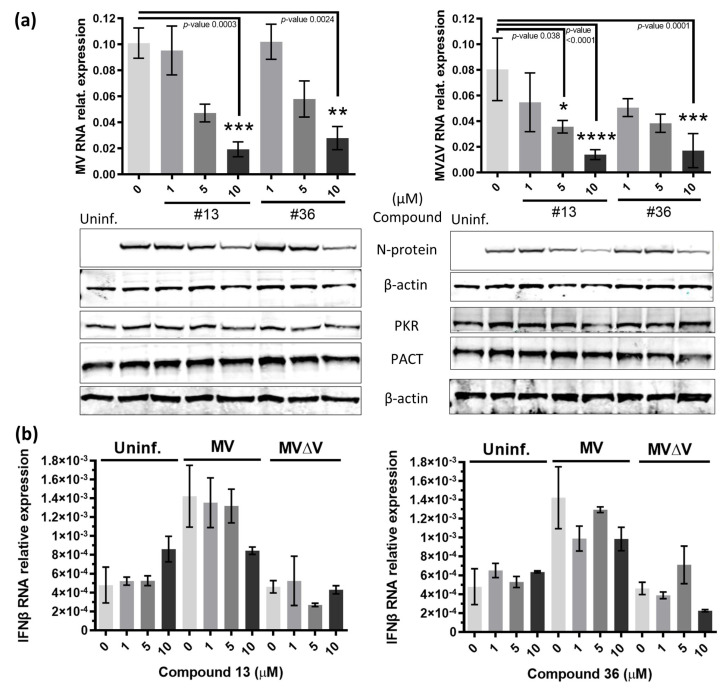
Effect of compounds **13** and **36** on measles virus (MV) infection. (**a**) After adsorption of uninfected or infected HEK293T cells with MV (left) or MVΔV (right), the medium was replaced with fresh medium containing the indicated concentrations of the compounds. After 24 h, total RNA was extracted for RT-qPCR detection of viral RNA genome, as in [30], as well as of (**b**) IFNβ mRNA. Total protein cell extracts were prepared for immunoblot (representative shown underneath the graph in (**a**)), to analyze the expression of the MV N protein and of endogenous PKR and PACT; detection of β-actin served as loading control. Averages of three technical replicates from two independent experiments are shown. For clarity, only statistically significant differences are indicated in (a), where both compounds inhibit both versions of MV at 10 µM after 24 h (*p*-values of 0.0003, 0024, <0.0001 and 0.0001, respectively), and compound **13** at 5 µM also inhibits MVΔV significantly (*p*-value 0.038). Statistical significance is indicated with asterisks, where * represents a *p*-value < 0.05, ** a *p*-value < 0.01, *** a *p*-value < 0.001, and **** a *p*-value < 0.0001.

**Figure 5 ijms-24-14791-f005:**
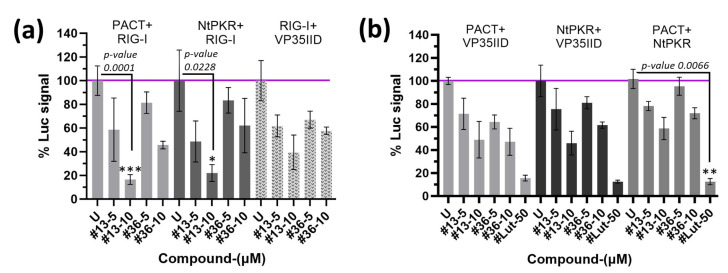
Effect of compounds **13**, **36** or luteolin on different protein pair interactions. HEK293T cells were co-transfected with 100 ng of the different constructs (PACT, RIG-I, NtPKR or VP35IID) bearing the luciferase moiety 1 or moiety 2 at N-terminus; luciferase signal is expressed as percent normalized to untreated (U) with compounds; the horizontal purple line indicates 100% luciferase signal. Representative experiments are shown; only statistically significant differences are indicated. (**a**) Effect of compounds **13** or **36** at different concentrations on RIG-I interaction with either PACT, NtPKR or VP35IID. (**b**) Effect of compounds **13**, **36** and luteolin on combinations of interacting PACT/NtPKR/VP35IID proteins. Representative graphs are shown; only statistically significant differences are indicated for clarity. PACT + RIG-I interaction is disrupted significatively (*p*-value 0.0001) by compounds **13** at 10 µM in comparison to U. NtPKR + RIG-I interaction is disrupted significatively (*p*-value 0.0228) by compounds **13** at 10 µM in comparison to U. PACT + NtPKR interaction is disrupted significatively (*p*-value 0.0066) by luteolin at 50 µM in comparison to U. Statistical significance is indicated with asterisks, where * represents a *p*-value < 0.05, ** a *p*-value < 0.01, and *** a *p*-value < 0.001.

**Figure 6 ijms-24-14791-f006:**
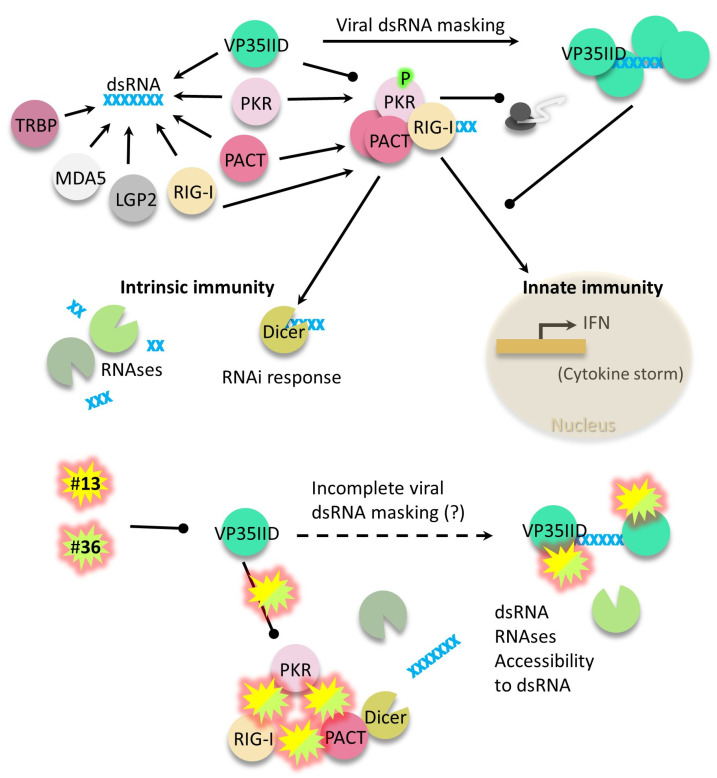
Summary model of the VP35IID crosstalk with dsRNA cell sensors and the effect of compounds #**13** and #**36**. Overall, these compounds restore intrinsic immunity response (immediate degradation of viral dsRNA by RNAses and RNAi) without affecting the inhibition of innate immunity by VP35IID, therefore preventing a cytokine storm aggravation. Trans-activation response (TAR) RNA-binding protein (TRBP); anti-melanoma differentiation-associated gene 5 (MDA5); Laboratory of Genetics and Physiology 2 (LGP2).

## Data Availability

This paper contains all relevant data required to support its results.

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
