# Peer review of "Novel Antiviral Molecules against Ebola Virus Infection"

_ijms, 2023, doi:10.3390/ijms241914791_

Round 1

Reviewer 1 Report

The manuscript entitled "Novel antiviral molecules against Ebola virus infection” by Rodriguez et al. describes the identified compounds as antiviral agents against EBOV and measles virus was addressed, as well as their effect on the intrinsic and innate immune response. In my opinion, the manuscript is need major revision to be suitable to publish in IJMS due to the following issues,

Please review all manuscript as there are several important typographical accentuation and grammatical errors throughout the paper.

There are many references in the introduction section that need to be updated, as they are more than 10 years old

Authors must follow the journal's guidelines because it is known that figures bearing the number S are in the appendix data, so all figures must be renumbered in order.

It does not logical for the authors to discuss the results and there is a separate discussion section, so all references from the results section must be moved to the discussion section

All the figures, it is necessary to indicate between whom the statistical comparisons are made, to mark them in the bars and to indicate them in the caption of the figure.

The morphological change of the cells must be studied to shows the interaction between the compounds 13 and 36 with EBOV and MV infection in cell culture using photographed with a confocal microscope in vitro.

There are several abbreviations in the M&M and in the results that need to be defined the first time they are written in the paper and their description is done later

The discussion section needs to be rewritten

There is no conclusion section

Minor editing of English language required

Author Response

Please review all manuscript as there are several important typographical accentuation and grammatical errors throughout the paper.

We thank the reviewer for pointing this. We have carefully re-read the manuscript and made appropriate corrections where needed.

There are many references in the introduction section that need to be updated, as they are more than 10 years old.

We would like to point out that we are indeed citing recent references (for instance, Jain, 2023; 2020; Yuan, 2020; Banerjee, 2020). As for references that may be considered as “older”, they belong to the rich history of this field (Ebola, VP35, dsRNA binding proteins) and we believe that they are of importance for those who are following that path of research.

Authors must follow the journal's guidelines because it is known that figures bearing the number S are in the appendix data, so all figures must be renumbered in order.

The reason why we have chosen to include the supplementary figures in the text, is because these data are immediately complementary to the main figures. Therefore, we did not want to disrupt the flow of reading by putting them in Appendixes. By doing this, we followed what was advised in the guidelines: https://www.mdpi.com/authors/layout#_bookmark83 "“Authors can use Appendixes to add further information to support the results reported in the manuscript. They should be used when including the information in the main text would disrupt the flow for readers or where only a minority of the audience is expected to be interested. (…)”.

It does not logical for the authors to discuss the results and there is a separate discussion section, so all references from the results section must be moved to the discussion section

We agree with the reviewer and we have now moved the section 2.1 paragraph starting from “Of note…” to the Discussion section. Some references are however kept in the result sections (for instance, in section 2.5) since they are required for the comprehension of the text.

All the figures, it is necessary to indicate between whom the statistical comparisons are made, to mark them in the bars and to indicate them in the caption of the figure.

The figures are described in the main text and all the necessary information is provided in the captions of the figure (statistical differences, p-values), whenever possible to ease the reading and avoid overcrowding of the figures. Figure S1, Figure 2b, Figure 4, and Figure 5 captions have been detailed as requested.

The morphological change of the cells must be studied to shows the interaction between the compounds 13 and 36 with EBOV and MV infection in cell culture using photographed with a confocal microscope in vitro.

Our initially designed high-throughput screening study aimed at finding novel compounds against EBOV. We also confirmed their antiviral potential in MV and extensively assessed their cytotoxicity. Therefore, showing “the interaction between the compounds 13 and 36 with EBOV and MV infection in cell culture using photographed with a confocal microscope in vitro” is out of scope of this study. Given their antiviral activity, we suggest to detail their action in animal models in future studies.

There are several abbreviations in the M&M and in the results that need to be defined the first time they are written in the paper and their description is done later

We have now re-read the manuscript carefully and added the required definitions where appropriate.

The discussion section needs to be rewritten

We have now made changes in the discussion section, such as including a section coming from the paragraph 2 of the results.

There is no conclusion section

We have followed the guidelines of the journal: “Conclusions: This section is not mandatory but can be added to the manuscript if the discussion is unusually long or complex.” https://www.mdpi.com/journal/ijms/instructions#suppmaterials We have considered that our manuscript is neither excessively long (< 5000 words) or complex (6 main figures) and the main message is well described in the abstract. Therefore, to avoid redundancy, we have preferred not to include a conclusion.

Comments on the Quality of English Language: Minor editing of English language required.

We have carefully re-read the manuscript and made the required corrections.

Reviewer 2 Report

The study identified two compounds that inhibit Ebola and measles virus infections in cells culture independently of IFN induction. This discovery has potential implications for antiviral treatments. They introduce the use of a chemical library screening approach, which is relatively novel in the context of EBOV research. The manuscript is generally well-written and clear. I believe this paper is publishable.

Author Response

We thank the reviewer for his/her nice and positive comments.

Reviewer 3 Report

In this manuscript, the authors screened 2P2I3D chemical library and found two compounds affect EBOV interaction. These two compounds disrupt the interaction of EBOV VP35IID with PKR and PACT and inhibit EBOV infection in cell culture and measles virus (MV) infection. The paper was nicely organized and written well. From my point of view, it is suitable for publication in "Journal of molecular Sciences" after the authors respond to the following question.

Specific comments are listed below:

1.      For all luciferase assay figures, please clarify the “Blank”, is empty vector or blank cells? Do authors normalize the Luc data? Transfection efficiency needs to be considered.

2.      Fig. 3 b and Fig.5, please do statistical analysis.

3.      They evaluate the inhibition of the two compounds in 293T and Huh7 cell. Do authors test them in primary cells? Function in primary cells and cell line sometime is different.

Author Response

  1. For all luciferase assay figures, please clarify the “Blank”, is empty vector or blank cells? Do authors normalize the Luc data? Transfection efficiency needs to be considered.

We thank the reviewer for noticing this point. In Figure 1b, 1c, and Figure S1, we have now added in the legend the explanation for the term “blank” which is noted on the abscissa of the graph “Blank shows very low levels of signal coming from the plastic plate without cells, media, DMSO or compounds to ensure the luciferase substrate does not give any signal by itself in comparison with the transfected samples.”

For Figure S2 (hits obtained from the screening), the procedure, including explanations for the controls which served to normalize the luc data, is described in Materials and Methods paragraph 4.5: “For each plate, the first four wells of column 1 and the four last wells of column 12 contained only 1 µL of DMSO to act as positive controls of fluorescence signal (cells co-transfected either with the VP35IID/NtPKR pair or the VP35IID/PACT pair). The remaining wells of columns 1 and 12 also contained 1 µL of DMSO but the added cells have been co-transfected with vectors expressing both luciferase moieties (Luc1 or Luc2) to serve as negative controls.” Luc data is either given as raw (absolute) values or normalised to DMSO when given in %, as indicated in the corresponding figure foots.

We have not strictly included a measure of the transfection efficiency in our experiments. However, our experiments were done in the most similar way as possible; same number of seeded cells, same time of treatment after seeding, used of the same batch of reagent for transfection, use of the same preparations of the different plasmids. The transfection protocol was also well established in the laboratory. In addition, internal controls (positive and negative controls) served to evaluate correctly the data.

  1. Fig. 3 b and Fig.5, please do statistical analysis.

Figure 3b. Given the limited accessibility to a BSL4 environment at the time of the study, we only have an n = 2 for each sample. Averages of them are shown on the histograms with their SD as indicated in section 4.10 of M&M. We indicate so on the figure “Graphs display averages of 2 technical replicates.” Since the sample is so small, we cannot do a confident statistical analysis, since any changes would not be significant. Therefore, we describe our observations in the main text only in a qualitative way:

“We evidenced that the presence of both intracellular and extracellular EBOV RNA was reduced by the compounds in a dose-dependent manner already 24 hpi (Figure 3b), and the production of the infectious virus (titre) qualitatively decreased between 0.5-1.5 logs (Figure 3c).”

Fig. 5. We thank the reviewer for pointing this. Statistical results are added onto the figure and explained in the figure foot.

  1. They evaluate the inhibition of the two compounds in 293T and Huh7 cell. Do authors test them in primary cells? Function in primary cells and cell line sometime is different.

We agree that in different cell lines (primary or not), the effect of the compounds may vary. We also agree that testing the compounds in primary cell lines as e.g., human fibroblasts could reinforce the findings. However, at the time of the study, we could not test the compounds in primary cells because we did not have relevant primary cell lines for both EBV and MV infections. We therefore used two widely used and different cell lines; one of embryonic origin (293T) and another transformed (Huh7). Nevertheless, we consider a systemic analysis of their effect in a rodent model may be even more suitable in future studies.

Round 2

Reviewer 1 Report

Thank you for making requested comments very carefully